# Mechanical Properties and Root Canal Shaping Ability of a Nickel–Titanium Rotary System for Minimally Invasive Endodontic Treatment: A Comparative In Vitro Study

**DOI:** 10.3390/ma15227929

**Published:** 2022-11-09

**Authors:** Hayate Unno, Arata Ebihara, Keiko Hirano, Yuka Kasuga, Satoshi Omori, Taro Nakatsukasa, Shunsuke Kimura, Keiichiro Maki, Takashi Okiji

**Affiliations:** Department of Pulp Biology and Endodontics, Division of Oral Health Sciences, Graduate School of Medical and Dental Sciences, Tokyo Medical and Dental University (TMDU), 1-5-45 Yushima, Bunkyo-ku, Tokyo 113-8549, Japan

**Keywords:** austenite, fatigue fractures, martensite, nickel–titanium, root canal preparation, torque

## Abstract

Selection of an appropriate nickel–titanium (NiTi) rotary system is important for minimally invasive endodontic treatment, which aims to preserve as much root canal dentin as possible. This study aimed to evaluate selected mechanical properties and the root canal shaping ability of TruNatomy (TRN), a NiTi rotary system designed for minimally invasive endodontic shaping, in comparison with existing instruments: HyFlex EDM (HEDM), ProTaper Next (PTN), and WaveOne Gold (WOG). Load values measured with a cantilever bending test were ranked as TRN < HEDM < WOG < PTN (*p* < 0.05). A dynamic cyclic fatigue test revealed that the number of cycles to fracture was ranked as HEDM > WOG > TRN > PTN (*p* < 0.05). Torque and vertical force generated during instrumentation of J-shaped artificial resin canals were measured using an automated instrumentation device connected to a torque and vertical force measuring system; TRN exhibited smaller torque and vertical force values in most comparisons with the other instruments. The canal centering ratio for TRN was smaller than or comparable to that for the other instruments except for WOG at the apex level. Under the present experimental conditions, TRN showed higher flexibility and lower torque and vertical force values than the other instruments.

## 1. Introduction

Nickel–titanium (NiTi) rotary instruments have gained popularity for root canal instrumentation because of their high flexibility [1] and ability to maintain root canal curvature [2,3]. In recent years, various technical advances in metallurgy, geometry, and kinematics have enabled the development of different NiTi rotary systems with improved efficiency and predictable safety [4]. In particular, heat treatment has been widely applied to NiTi alloys to improve the mechanical characteristics of NiTi rotary instruments, including flexibility and resistance to cyclic fatigue, by adjusting phase transition temperatures to induce the growth of the ductile martensite phase and R-phase [5]. For these newly introduced NiTi rotary instruments, scientific evidence regarding preclinical parameters, such as root canal shaping ability and fracture resistance for their appropriate clinical application, should be gathered.

ProTaper Next (PTN; Dentsply Sirona, Ballaigues, Switzerland) is a well-investigated heat-treated NiTi rotary system made of M-wire with improved cyclic fatigue resistance [6]. PTN has an off-centered square cross-section with a variable taper [7]. HyFlex EDM (HEDM; Coltène-Whaledent, Altstätten, Switzerland) is manufactured from distinctively heat-treated CM-wire, together with electrical discharge machining [8], and shows improved cutting efficiency and cyclic fatigue resistance [9]. The cross-section of HEDM is variable: rectangular, trapezoidal and triangular in the apical, middle, and coronal portions, respectively. WaveOne Gold (WOG; Dentsply Sirona, Ballaigues, Switzerland) is a representative NiTi reciprocating system manufactured from Gold-wire with a parallelogram cross-sectional shape with two cutting edges. WOG reportedly exhibits higher flexibility and cyclic fatigue resistance [10], and a comparable ability to maintain canal curvature [11] as its predecessor, WaveOne (Dentsply Sirona). 

The concept of minimally invasive endodontic treatment involves preserving as much of the root dentin as possible [12]. The use of instruments with a small diameter and taper is advocated to limit the amount of dentin cutting and preserve the cervical dentin [13] to improve resistance to tooth fracture [14]. TruNatomy (TRN; Dentsply Sirona) is a recently launched NiTi rotary system with this concept incorporated into its design features. TRN is produced from a proprietary post-manufacture heat-treated NiTi alloy with a regressive taper, maximum flute diameter of 0.8 mm, and a parallelogram, off-centered cross-sectional design [15,16,17]. The TRN system consists of an orifice shaping file, a glide path file, and three files for root canal shaping with a single-length technique. TRN is reported to be more resistant to cyclic fatigue in comparison with instruments made of differently heat-treated alloys, such as PTN [17] and Vortex Blue (Dentsply Sirona) [16]. The degree of apical canal deviation induced by TRN is smaller than [18] or similar to [19] several other NiTi rotary systems. 

More information is, however, required to determine what advantages TRN offers over other NiTi rotary and reciprocating systems for performing minimally invasive endodontic shaping. Therefore, with the aim of evaluating the properties of TRN, this study examined this instrument’s flexibility, resistance to cyclic fatigue, generation of torque and vertical force during instrumentation, and root canal shaping ability, in comparison with those of other contemporary NiTi rotary and reciprocating systems. The null hypothesis was that there are no significant differences in bending load, dynamic cyclic fatigue resistance, torque and vertical force values during canal instrumentation, and root canal centering ability among the tested instruments.

## 2. Materials and Methods

Ethical approval is not applicable to this study because it does not contain any studies with human or animal subjects/materials.

### 2.1. Sample Size Estimation

G*Power software (version 3.1.9.7; Heinrich Heine Universität, Düsseldorf, Germany) was used to determine the sample size required. A priori analysis of variance (ANOVA; fixed effects, omnibus, one-way) was selected from the F-test family. The effect size was set at 0.6 based on data from a previous study [20]. An alpha-type error of 0.05 and a power beta of 0.85 were also specified. The required sample size of 10 per group was obtained.

### 2.2. Bending Test

TRN Prime (#26/0.04 taper, Dentsply Sirona), HEDM One File (#25/0.08-0.04 taper, Coltène-Whaledent), PTN X2 (#25/0.06-0.07 taper, Dentsply Sirona), and WOG Primary (#25/0.07-0.03 taper, Dentsply Sirona) were tested, using a self-made cantilever bending tester described in previous studies [21,22] (n = 10 for each model) (Figure 1). The file was fixed at 7 mm from the tip, and loaded at a position 2 mm from the tip at a speed of 1.0 mm/min until there was 3 mm of displacement. The bending load was measured at 0.5 mm and 2.0 mm of displacement. All experiments were performed at room temperature.

### 2.3. Dynamic Cyclic Fatigue Test

The same instruments as in the bending test were evaluated. A self-made cyclic fatigue tester with a movable test stand (MH2-500N; IMADA, Aichi, Japan) and the X-Smart Plus endodontic motor (Dentsply Sirona) were used [23] (Figure 2). An artificial root canal made of stainless steel and designed with a 1.5 mm diameter, a 60° curvature, 3 mm radius of curvature, and the center of the curvature at 5 mm from the tip of the instrument was used [24]. The instruments were rotated as per the manufacturers’ recommendations (500 rpm, 1.5 N·cm in TRN; 400 rpm, 2.4 N·cm in HEDM; 300 rpm, 2.0 N·cm in PTN; and the WaveOne Gold setting in WOG), while moving the handpiece with an axial up-and-down motion of 2 mm amplitude at 300 mm/min. The canal was lubricated with silicon oil (KF-96-100CS, Shin-Etsu Chemical, Tokyo, Japan). The length of time to fracture was measured, and the number of cycles to failure (NCF) was determined as the number of revolutions (rpm) × time to fracture (seconds). All experiments were performed at room temperature.

### 2.4. Measurement of Torque and Vertical Force

A self-made automated root canal instrumentation device connected to a torque and vertical force measuring system was employed as described in previous studies [20,25,26,27] (Figure 3). Briefly, this device consisted of the X-Smart Plus motor and a test stand (MX2-500N, IMADA, Aichi, Japan), and the torque and vertical force measuring system was configured with a built-in load cell (LUX-B-ID, Kyowa, Tokyo, Japan) for determining vertical force, and strain gauges (KFG-2-120-D31-11, Kyowa, Tokyo, Japan) for determining torque. The movable device of the stand, to which the handpiece was attached, was programmed to make an up-and-down motion at a speed of 50 mm/min for 2 s downward and 1 s upward [25,28].

J-shaped artificial resin root canals (0.02 taper, 0.1 mm apical foramen diameter, 45° curvature, 16 mm full working length; Endo Training Bloc, Dentsply Sirona; n = 40) were randomly assigned to Groups TRN, HEDM, PTN and WOG (n = 10, each) and instrumented according to the sequence recommended for narrow canals by the manufacturers. After coronal flaring to a depth of 12 mm, the canal was subjected to automated instrumentation consisting of glide path preparation to the full working length (16 mm), followed by a two-instrument full-length instrumentation where the working lengths were set incrementally at 14, 15, and 16 mm for the two instruments. After each instrumentation, a #10 K-file (Zipperer, Munich, Germany) was used to verify the patency, and canal irrigation was performed with 1 mL of distilled water. The canal was lubricated with RC-Prep (Premier Dental, Plymouth Meeting, PA, USA). Torque (cutting direction) and vertical force (downward and upward) were monitored during the automated root canal instrumentation and the maximum values were statistically analyzed.

The rotary instruments used were as follows: TRN Orifice Modifier (#20/0.08 taper), TRN Glider (#17/0.02 taper), TRN Small (#20/0.04 taper) and TRN Prime (#26/0.04 taper) in Group TRN; HEDM Orifice Opener (#25/0.12 taper), HEDM Glide Path File (#10/0.05 taper), HEDM Preparation File (#20/0.05 tape) and HEDM One File (#25/0.08–0.04 taper) in Group HEDM; ProTaper SX (#19/0.04 taper; Dentsply Sirona), ProGlider (#16/0.02 taper; Dentsply Sirona), PTN X1 (#17/0.04 taper) and PTN X2 (#25/0.06 taper) in Group PTN; and ProTaper Gold SX (#19/0.04 taper), WOG Glider (#15/0.06 taper), WOG Small (#20/0.07 taper), and WOG Prime (#25/0.07 taper) in Group WOG. The rotational speeds and torque-limit values were set as per the manufacturers’ recommendations as follows: 500 rpm, 1.5 N·cm for TRN; 300 rpm, 1.8 N·cm for HEDM Glide Path File; 400 rpm, 2.4 N·cm for HEDM; 250 rpm, 4.0 N·cm for ProTaper SX; 300 rpm, 2.0 N·cm for PTN; 300 rpm, 3.0 N·cm for ProTaper Gold SX; and the WOG mode for WOG. All experiments were performed at room temperature.

### 2.5. Root Canal Centering Ability

Preoperative (before the glide path preparation) and postoperative digital images of the artificial resin canals described above were obtained using a digital microscope (VH-8000, Keyence, Osaka, Japan). The images were then overlaid and analyzed with image analysis software (Adobe Photoshop Elements 2021, San Jose, CA, USA) [28]. The canal centering ratio was calculated at 0, 0.5, 1, 2, and 3 mm from the apex using the following formula: (X–Y)/Z
where:

X = amount of removed resin material from the outer canal wall

Y = amount of removed resin material from the inner canal wall

Z = root canal diameter after instrumentation [29].

The post-instrumentation canal deviation becomes smaller as the centering ratio approaches 0. All instrumentation was performed at room temperature.

### 2.6. Statistical Analysis

SPSS software (version 27.0; IBM, Armonk, NY, USA) was used to determine statistical differences at a significance level of 5%. Data normality and homogeneous variance were verified using the Shapiro–Wilk test and the Levene’s F test, respectively. The values obtained in the bending test, torque and vertical force values, and the canal centering ratio were analyzed with two-way ANOVA and the Tukey post-hoc test. The NCF values were analyzed by one-way ANOVA and the Tukey post-hoc test.

## 3. Results

### 3.1. Bending Loads

At a 0.5 mm deflection, PTN withstood a significantly larger load than the other instruments (*p* < 0.05; Figure 4). No significant difference was observed among TRN, HEDM, and WOG (*p* > 0.05). At a 2.0 mm deflection, the load values were ranked as TRN < HEDM < WOG < PTN (*p* < 0.05; Figure 4).

### 3.2. Dynamic Cyclic Fatigue Resistance

The NCF value was the highest in HEDM, followed by WOG, TRN, and PTN (*p* < 0.05; Figure 5).

### 3.3. Torque and Vertical Force

Figure 6 shows the maximum torque and vertical force values produced by each instrument in each group. No file fracture occurred during root canal instrumentation.

Upward force values for the first shaping instruments were ranked as HEDM < TRN < WOG < PTN (*p* < 0.05). Among the second shaping instruments, TRN showed a significantly lower upward force compared with the other instruments (*p* < 0.05). Intragroup comparisons revealed that all shaping instruments, except the first instrument in HEDM, showed significantly higher values than the corresponding glide path instrument.

Regarding the downward force, WOG demonstrated significantly higher values than TRN (*p* < 0.05) when the first shaping instruments were compared. Among the second shaping instruments, PTN and HEDM exhibited significantly higher values than the other instruments (*p* < 0.05). Intragroup comparisons revealed that the first shaping instrument in WOG and the second shaping instrument in HEDM and PTN exhibited significantly higher values than the corresponding glide path instruments (*p* < 0.05), while the three instruments in Group TRN showed similar values.

Torque values in the cutting direction among the first shaping instruments were ranked as PTN and WOG > TRN and HEDM (*p* < 0.05). Torque values among the second shaping instruments were TRN < PTN < HEDM and WOG. Within Group HEDM, Group PTN, and Group WOG, torque values were ranked as glide path instrument < first shaping instrument < second shaping instrument (*p* < 0.05). Within Group TRN, the first shaping instrument generated a significantly higher torque value than the glide path instrument (*p* < 0.05).

### 3.4. Canal Centering Ratio

At the 0 mm level, Group WOG showed a significantly smaller value than the other groups (*p* < 0.05; Figure 7). At the 0.5 mm level, significantly higher and lower values were found in Group WOG and Group PTN, respectively (*p* < 0.05; Figure 7). At the 1 mm and 2 mm levels, the ratios of Group PTN were significantly higher than those of the other groups (*p* < 0.05; Figure 7). No significant differences were found among the groups at 3 mm from the apex.

## 4. Discussion

The aim of minimally invasive endodontic treatment is to minimize the amount of dentin cutting, particularly near the cervical area [17,19]. However, restricted coronal enlargement in a curved canal may make an instrument more stressed and the apical portion of a canal more prone to deviation during instrumentation [19,30]. Thus, selection of appropriate instruments with sufficient fracture resistance and canal centering ability is important to avoid iatrogenic events.

In this study, a series of experiments testing flexibility, resistance to cyclic fatigue, torque and vertical force generation, and root canal shaping ability were conducted in an attempt to obtain a comprehensive understanding of the characteristics of TRN as a NiTi rotary system suitable for minimally invasive endodontic treatment. The results demonstrated significant differences in all the tests across the tested instruments; in particular, TRN exhibited a significantly smaller bending load, and significantly lower torque and vertical force values in several comparisons with the other instruments. Thus, the null hypothesis was rejected.

Various factors such as geometry [31], heat treatment [1,32], and the alloy manufacturing process [5,33] affect the mechanical properties of NiTi rotary instruments. Because no existing NiTi rotary systems are identical to TRN in terms of overall design, control instruments were selected from dimensionally similar, well-studied systems, as in several other studies [17,19,34,35,36]. This made the comparisons sufficiently meaningful to test the performance of TRN in comparison with instruments with different metallurgy and rotational modes.

In the bending test, the displacement at 0.5 mm corresponds to the elastic region, and at 2.0 mm to the superelastic region [37]. In this study, TRN exhibited the smallest bending load in the superelastic region, which corroborates the high flexibility of TRN as reported in a previous study [38]. Several factors influence the flexibility of NiTi rotary instruments, including geometry (e.g., size, cross-sectional design, core diameter, and pitch length) [39,40] and metallurgy [5,33]. Regarding metallurgy, differential scanning calorimetric studies have reported that the austenite starting and finishing temperatures of TRN are 11.8 °C and 29.2 °C, respectively [38], in contrast with those reported for PTN (0.2 °C and 51.4 °C; [41]), HEDM (45.47 °C and 51.55 °C; [42]), and WOG (8.5 °C and 51.6 °C; [41]). Thus, at the tested temperature, TRN was in the mixed phase of austenite, R-phase, and martensite, and HEDM was in the highest martensite composition, which indicates that the lowest flexibility of TRN cannot be explained by its phase composition. Thus, the highest flexibility of TRN may be primarily attributable to the geometric factors of TRN, such as the smaller wire diameter (0.8 mm in TRN vs. 1.2 mm in the other instruments) [17], the parallelogram cross-sectional design, which may contribute to decreasing the core diameter [40], and the smallest taper in the tip region.

The dynamic cyclic fatigue test incorporates axial movement mimicking a pecking motion and thus better simulates clinical conditions compared to the static cyclic fatigue test [43,44,45,46]. In the dynamic test, the fatigue life of NiTi instruments is longer than that in the static test because the axial movement releases the stress concentrated at the curvature [43,44,47]. In this study, NCF values were ranked as PTN < TRN < WOG < HEDM, which can be attributed to differences in several factors, including heat treatment, kinematic movement, and cross-sectional design [48]. The NCF value for TRN was significantly higher than that for PTN, which is consistent with a previous study [17]. TRN and PTN differ in heat treatment (post- vs. pre-manufacturing treatment), taper, and cross-sectional design (parallelogram vs. rectangle), while it is unclear which of these factors contribute more to the higher NCF of TRN than PTN. Of interest was that the NCF of TRN was lower than that of HEDM and WOG, which is contradictory to the notion that highly flexible instruments yield higher cyclic fatigue resistance. Additionally, the present results do not agree with the findings that instruments with a smaller diameter perform better in the cyclic fatigue test [49]. These contradictory results may be explained by the fact that the martensite-rich metallurgy in HEDM [48,50] and the reciprocal motion in WOG [51] counteracted the impact of the diameter and flexibility, resulting in the higher cyclic fatigue life of these instruments.

NiTi rotary instruments exert torque and downward force to cut dentin, and if the torsional stress accumulated in the instruments exceeds the elastic limit, torsional fracture may occur [52]. The upward force represents the screw-in force, which may expose a rotating file to a risk of sudden torsional stress leading to separation [29,31,53]. In this study, the torque and force produced by all the glide path instruments were low. Among the shaping instruments, TRN in general produced lower torque and vertical force than the other instruments. These findings are consistent with a previous finding that TRN produced smaller torque and vertical force than PTN [17], and may be attributed to the cumulative effect of various influencing factors. Smaller taper and diameter [54] and higher flexibility [31,55] are among detrimental factors associated with lower torque/force production. The off-centered parallelogram cross-sectional design of TRN, which contacts with the root canal wall at one or two points during instrumentation, may also play a vital role in its lower screw-in force generation [53]. In addition, pre-instrumentation canal size/taper relative to the dimension of an instrument may have an impact on torque/force generation. This may explain the significant increase, compared with the previous instrument, of torque and/or upward force in the first shaping instrument in TRN, PTN, and WOG and the second shaping instrument in HEDM and PTN; these instruments are greater in both size and taper compared to the previous instrument. These finding imply which instrument(s) are stressed, and thus may clinically require careful manipulation during the instrumentation sequence of each system.

The canal centering ratio demonstrated for TRN was, in general, smaller than or comparable to that for the other instruments at each measuring point, which may be explained by the fact that flexible instruments cause less canal deviation [56,57]. At the 0 mm point, however, WOG showed a significantly smaller ratio than TRN. This may be due to the reciprocating motion of WOG, which disengages a bound instrument during rotation in the non-cutting direction and thus reduces the contact time of the blades to the canal wall, leading to improved canal centering ability [58].

Collectively, the present findings suggest that TRN has several mechanical properties that favor its use under the concept of minimally invasive endodontic treatment, such as higher flexibility and lower torque/force values than the other instruments. However, this study is not without limitations. One limitation is the use of artificial resin canals to evaluate torque and vertical force generation and shaping ability; although resin canals offer a highly standardized root canal morphology, their hardness is lower and their surface smoother than those of real canals and thus less force may be required for instrumentation [59]. Another potential limitation is the test temperature (room temperature), under which the metallurgical microstructure and mechanical properties of NiTi instruments may be different from those under clinical conditions [38]. Accurate temperature measurement of NiTi instruments during root canal shaping under clinical conditions has not yet been reported; in future experiments, temperature taken in real time should be taken into account. Further study is required to determine how NiTi instruments perform differently under different temperature settings between the laboratory and clinical conditions. Incorporation of tests for other relevant parameters, such as torsional resistance, surface hardness, and cutting efficiency, may facilitate more comprehensive analysis. Future studies should also be carried out to evaluate the influence of the root canal anatomy, which is highly variable [60] with several factors, including degree of curvature, that may influence cyclic fatigue resistance [24] and torque and force generation [52] of NiTi rotary instruments. Increasing knowledge of the characteristics of NiTi rotary instruments will be beneficial for clinicians in making better selection of these instruments according to individual cases.

## 5. Conclusions

This study examined the flexibility, resistance to cyclic fatigue, generation of torque and vertical force during instrumentation, and root canal shaping ability of TRN, in comparison with those of HEDM, PTN, and WOG. The following conclusions were drawn:The bending load values were not significantly different between TRN, HEDM, and WOG at a 0.5 mm deflection. At a 2.0 mm deflection, the load values were ranked as TRN < HEDM < WOG < PTN.The number of cycles to fracture was ranked as HEDM > WOG > TRN > PTN.During instrumentation of J-shaped artificial resin canals, TRN showed smaller torque and vertical force values in most comparisons with the other instruments.The canal centering ratio for TRN was smaller than or comparable to that for the other instruments, except for WOG at the apex level.TRN showed higher flexibility and lower torque/force values than the other instruments, which may favor its clinical use under the concept of minimally invasive endodontic treatment.

## Figures and Tables

**Figure 1 materials-15-07929-f001:**
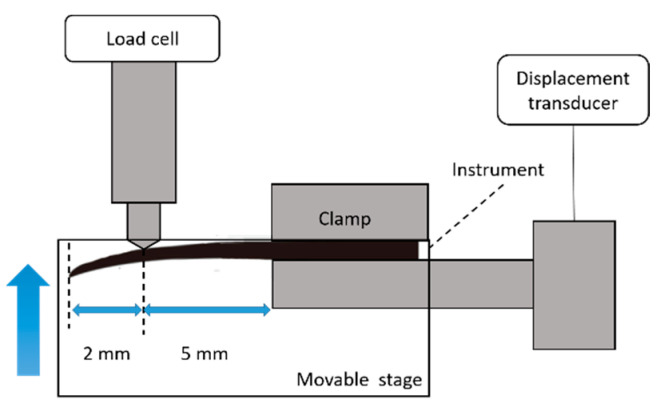
Schematic drawing showing the cantilever bending tester used in the experiment.

**Figure 2 materials-15-07929-f002:**
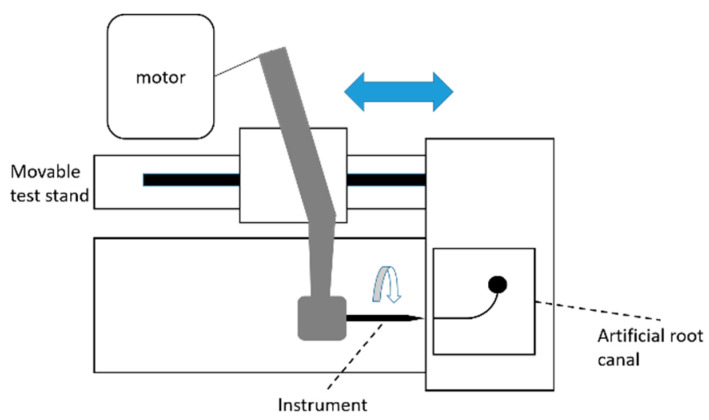
Schematic drawing showing the cyclic fatigue tester used in the experiment.

**Figure 3 materials-15-07929-f003:**
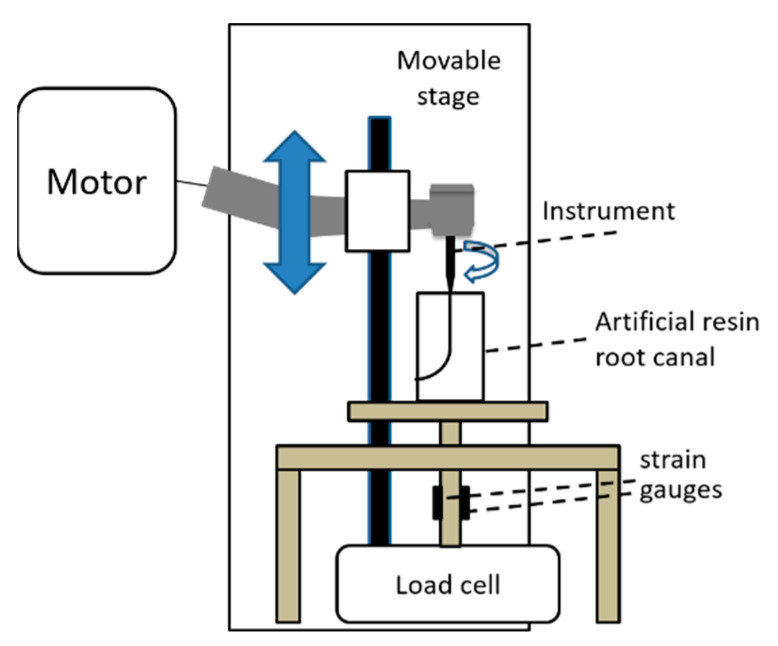
Schematic drawing of the automated root canal instrumentation device and the torque and vertical force measuring system used in the experiment.

**Figure 4 materials-15-07929-f004:**
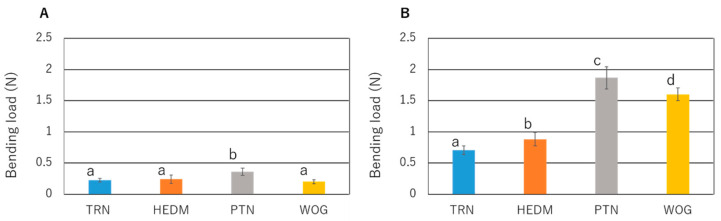
Bending load values (N) of the tested NiTi rotary instruments at a deflection of 0.5 mm (elastic region: **A**) and 2.0 mm (superelastic region: **B**). Values are means and standard deviations (n = 10). Groups with different letters in each testing condition are significantly different (*p* < 0.05; one-way ANOVA and Tukey test).

**Figure 5 materials-15-07929-f005:**
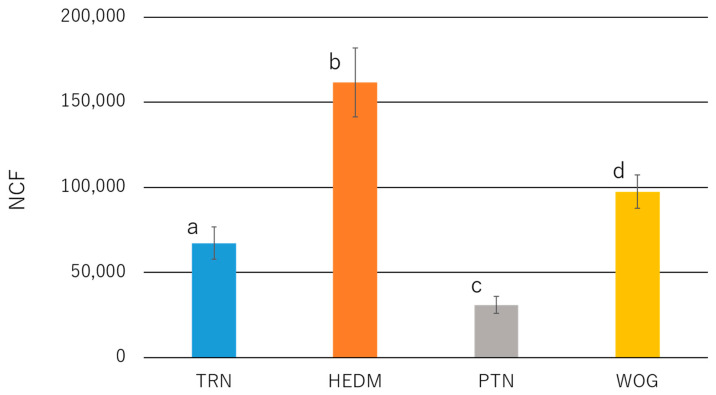
Number of cycles to failure (NCF) of NiTi rotary instruments subjected to a dynamic cyclic fatigue test. Values are means and standard deviations (n = 10). Groups with different letters are significantly different (*p* < 0.05; one-way ANOVA and Tukey test).

**Figure 6 materials-15-07929-f006:**
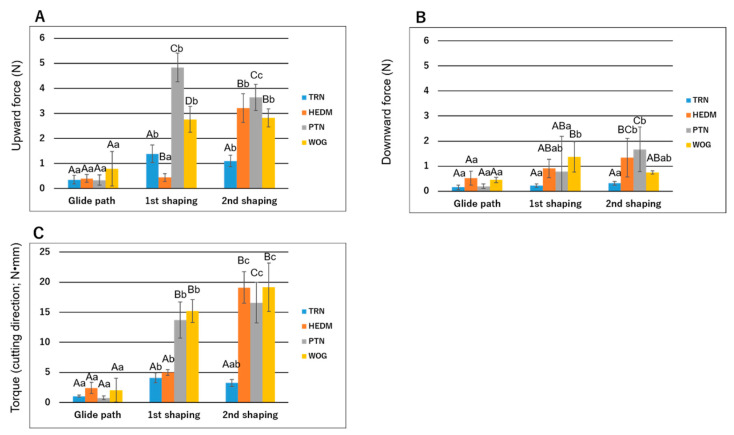
Maximum upward force (**A**), downward force (**B**), and torque in the cutting direction (**C**) generated during root canal instrumentation. Values are means and standard deviations (n = 10). Different uppercase letters in each instrumentation sequence in each panel indicate that the values are significantly different in intergroup comparisons (*p* < 0.05; two-way ANOVA and Tukey test). Different lowercase letters in each panel indicate that the values are significantly different within the same instrument (*p* < 0.05; two-way ANOVA and Tukey test).

**Figure 7 materials-15-07929-f007:**
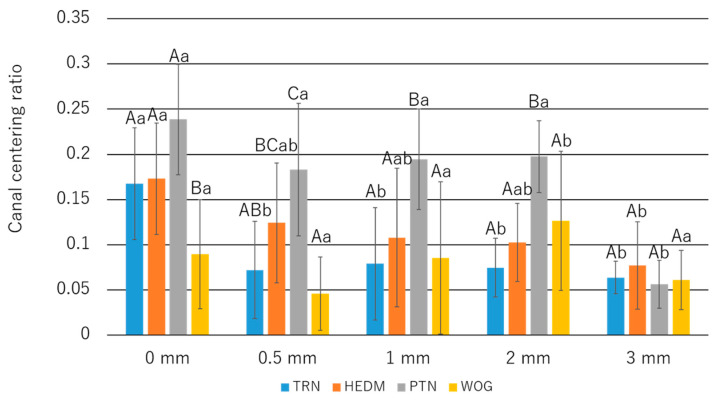
Canal centering ratios. Values are means and standard deviations (n = 10). Different uppercase letters in the measurement level indicate that the values are significantly different in intergroup comparisons (*p* < 0.05; two-way ANOVA and Tukey test). Different lowercase letters indicate that the values are significantly different within the same instrument (*p* < 0.05; two-way ANOVA and Tukey test).

## Data Availability

The data presented in this study are available on request from the corresponding author.

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
