# Peer review of "Mechanical Properties and Root Canal Shaping Ability of a Nickel–Titanium Rotary System for Minimally Invasive Endodontic Treatment: A Comparative In Vitro Study"

_materials, 2022, doi:10.3390/ma15227929_

Round 1

Reviewer 1 Report

The paper entitled “Mechanical Properties and Root Canal Shaping Ability of aNickel-Titanium Rotary System for Minimally Invasive Endodontic Treatment” is a contribute that  aims evaluate selected mechanical properties and the root canal shaping ability of TruNatomy (TRN). The work  did not provides relevant original data but it could be considered anyway of interest for the readers. However before it could be considered valid for publication requires some corrections.

INTRODUCTION

Overall well structured, it provides all the information necessary to understand the scientific background, the knowledge gap and the objectives of the study.

MATERIAL AND METHODS

The scientific methodology used were described in a clear and exhaustive manner. There is no iconography related to the methodology used in the research. In this part of the paper should be added some figures. The statistical analysis used to evaluate the results is correct.

RESULTS AND

The results are described in a precise and detailed manner; graphical representation is well executed and allows a faster understanding of the results achieved in the study.

DISCUSSION

The discussion of the results is on the whole well articulated ; clinical relevance of the results should be emphasized more.

CONCLUSION

Conclusions are limited to a synthetic summary of the results obtained; this section must be revised and report preferably with a bulleted list, only the key results of the study.

Reviewer 2 Report

A review report of the manuscript titled "Mechanical Properties and Root Canal Shaping Ability of a Nickel-Titanium Rotary System for Minimally Invasive Endo-3 dontic Treatment". Authors of current study aimed to evaluate selected mechanical properties and the root canal shaping ability of TruNatomy (TRN), a NiTi rotary system designed for minimally invasive endodontic shaping, in comparison with existing instruments: HyFlex EDM (HEDM), ProTaper Next (PTN), and WaveOne 18 Gold (WOG). They cancluded that TRN has higher flexibility and lower torque and vertical force values than the other instruments.

Here are my concerns, questions and recommendations:

1. I could not understand why authors did not include control group for better comparison and interpretation of results and conclusion?

2.Line 92-93. Correct the sentance.

3.Regarding the root shapes, I suggest to add recent studies 1.     Mirza MB, Gufran K, Alhabib O, Alafraa O, Alzahrani F, Abuelqomsan MS, Karobari MI, Alnajei A, Afroz MM, Akram SM, Heboyan A. CBCT based study to analyze and classify root canal morphology of maxillary molars - A retrospective study. Eur Rev Med Pharmacol Sci. 2022 Sep;26(18):6550-6560. doi: 10.26355/eurrev_202209_29753. PMID: 36196703.

4.Authors should clearly highlight the clinical significance of the study.

5. In the end of the Discussion, future research suggestions should be provided.

6. References should be corrected according to journal requirements.

Overall article is well written, but in my view it can be considered for possible publication after thorough revision and clarification of concerns.

Reviewer 3 Report

This study aimed to evaluate selected mechanical properties and the root canal shaping ability of 16 TruNatomy (TRN), a NiTi rotary system designed for minimally invasive endodontic shaping, in 17 comparison with existing instruments: HyFlex EDM (HEDM), ProTaper Next (PTN), and WaveOne 18 Gold (WOG).

In general, the manuscript is well-written and the topic is important in the endodontic field. However, I have some considerations in order to improve the quality of this manuscript:

1.     Please review that the keywords are MeSH terms. I am not sure that these keywords should be adequate

2.     Please clarify the type of study or design (the study should be named appropriately).

3.     Please mention the ethical consideration, where applicable. If not, explain why not (in the method section). Ethical issues are important and these considerations should take into account the responsibility of the authors with the study. I have some concerns, because the manuscript did not have Institutional Review Board Statement

Why is Not applicable?

4.     The manuscript should consider the main guidelines to report preclinical studies (as the authors do not clearly mention the type of study design). Please review this reference:

Faggion CM Jr. Guidelines for reporting pre-clinical in vitro studies on dental materials. J Evid Based Dent Pract. 2012 Dec;12(4):182-9. doi: 10.1016/j.jebdp.2012.10.001. PMID: 23177493.

The authors should verify the accomplishment of these guidelines and attach a file containing the report of the items as proposed. The authors could use other kinds of convenient guidelines. This is important in order to obtain methodological quality.

5.     Please mention recommendations for research and practice (clinical applicability)

By following these recommendations, the manuscript could be accepted in this journal.

Have a good day

Round 2

Reviewer 2 Report

Authors have revised the manuscrip according to my comments and improved the manuscript. I do believe that manuscript suitable for publication in its current form. 

Reviewer 3 Report

Accepted